Analysing transcriptomic signatures and identifying potential genes for the protective effect of inactivated COVID-19 vaccines

Chen Hongquan 1 2
Zhang Lu 1 2
Xu Chen 1 2
Shen Xiaoyun 3
Lou Jiazhou 1 2
Wu Shengjun 3202107@zju.edu.cn 1 2
1 Department of Clinical Laboratory, Sir Run Run Shaw Hospital, Zhejiang University School of Medicine, Zhejiang University , Hangzhou , China
2 Key Laboratory of Precision Medicine in Diagnosis and Monitoring Research of Zhejiang Province, Zhejiang University , Hangzhou , China
3 Key Laboratory of Endoscopic Technology Research, Sir Run Run Shaw Hospital, Zhejiang University School of Medicine, Zhejiang University , Hangzhou , China
Zhan Cheng
Electronic publication date: 2023 Apr 19
Publication date: 2023
Volume: 11
Electronic Location ID: e15155
Received 2023 Feb 2; Accepted 2023 Mar 10
Copyright: ©2023 Chen et al.
Copyright year: 2023
Copyright holder: Chen et al.
License: This is an open access article distributed under the terms of the Creative Commons Attribution License, which permits unrestricted use, distribution, reproduction and adaptation in any medium and for any purpose provided that it is properly attributed. For attribution, the original author(s), title, publication source (PeerJ) and either DOI or URL of the article must be cited.
License URL: https://creativecommons.org/licenses/by/4.0/

Keywords: SARS-CoV-2, Inactivated vaccine, RNA-seq, Transcriptome profile, Neutralization antibody

Funding: Department of Education of Zhejiang Province Y202043596 Natural Science Foundation of Zhejiang province LQ21H190005 National Natural Science Foundation of China 82102476 This study was funded by Department of Education of Zhejiang Province (Y202043596), Natural Science Foundation of Zhejiang province (LQ21H190005) and National Natural Science Foundation of China (82102476). There was no additional external funding received for this study. The funders had no role in study design, data collection and analysis, decision to publish, or preparation of the manuscript.

==============================
Inactivated vaccines are one of the most effective strategies for controlling the coronavirus disease 2019 (COVID-19) pandemic. However, the response genes for the protective effect of inactivated vaccines are still unclear. Herein, we analysed the neutralization antibody responses elicited by vaccine serum and carried out transcriptome sequencing of RNAs isolated from the PBMCs of 29 medical staff receiving two doses of the CoronaVac vaccine. The results showed that SARS-CoV-2 neutralization antibody titers varied considerably among individuals, and revealed that many innate immune pathways were activated after vaccination. Furthermore, the blue module revealed that NRAS, YWHAB, SMARCA5, PPP1CC and CDC5L may be correlated with the protective effect of the inactivated vaccine. Additionally, MAPK1, CDC42, PPP2CA, EP300, YWHAZ and NRAS were demonstrated as the hub genes having a significant association with vaccines. These findings provide a basis for understanding the molecular mechanism of the host immune response induced by inactivated vaccines.

Introduction

According to a World Health Organization (WHO) report, the outbreak of severe acute respiratory syndrome coronavirus 2 (SARS-CoV-2) has caused a global pandemic with millions of deaths. The infected individuals have been reported to range from no clinical manifestation to a series of symptoms including fever, cough, sore throat, and shortness of breath and may even develop acute respiratory distress syndrome (ARDS) in severe cases (Chen et al., 2020; Huang et al., 2020; Wu et al., 2020). As a respiratory disease, SARS-CoV-2 appears to undergo rapid transmission (Ike et al., 2019; Ruan et al., 2021), which results in considerable economic costs and impacts on human health around the world. Controlling measures are urgently needed.

To date, no effective drugs or optimal treatments for SARS-CoV-2 infections have been developed (Abu-Raddad et al., 2021; Fernandes et al., 2022; He et al., 2022; Mistry et al., 2021). Fortunately, researchers and physicians have developed a series of vaccines for SARS-CoV-2 with different technical routes (Ali et al., 2021; Arunachalam et al., 2021; Ewer et al., 2021; Tauzin et al., 2022; Vikkurthi et al., 2022; Zhang et al., 2021c), including inactivated whole-virion vaccines (e.g., CoronaVac), RNA-based vaccines (e.g., BNT162b2 and mRNA-1273), and nonreplicating viral vector vaccines (e.g., AZD1222). The successful development of vaccines has greatly boosted confidence in the ability to stop the coronavirus disease 2019 (COVID-19) epidemic. Recent investigations have suggested that these vaccines can induce effective immune responses among recipients with few untowards effects and greatly reduce the clinical symptoms and mortality risk of infected patients (Tanriover et al., 2021; Vikkurthi et al., 2022). Researchers revealed that the BBIBP-CorV vaccine activated innate and adaptive immune cells at the transcriptional level after vaccination (Yin et al., 2023). However, the host immune response to SARS-CoV-2 inactivated vaccines at the transcriptional level are not yet fully understood. In addition, due to rapid evolution of the virus, the epidemic continues to evolve dramatically worldwide (Le et al., 2022; Rana et al., 2022), and a more comprehensive understanding of the underlying immune response to and gene network regulation in response to the vaccines we presently use is urgently needed. These insights may provide a basis for understanding the molecular mechanism of the host immune response induced by inactivated vaccines.

In this study, RNA sequencing techniques were used to identify changes in transcriptomes using peripheral blood mononuclear cell (PBMC) specimens collected from 29 medical staff members on Day 21 after receiving the second dose of CoronaVac, and the transcriptomic data of three infected patients and three healthy individuals were downloaded from a published paper as controls (Xiong et al., 2020). Differential gene expression analysis revealed several key pathways correlated with the immune response. In addition, key modules that are strongly associated with clinical traits of vaccination were identified using the weighted correlation network analysis (WGCNA) method. Moreover, NRAS, YWHAB, SMARCA5, PPP1CC and CDC5L were the shared response genes in the blue module and were significantly downregulated in vaccinated individuals compared with those of infected patients. In addition, MAPK1, CDC42, PPP2CA, EP300, YWHAZ and NRAS were showed to be hub genes in the blue module that had a significant association with vaccines.

Materials & Methods

Study population

A cohort of 29 medical staff were recruited at Sir Run Run Shaw Hospital, Zhejiang University School of Medicine, Hangzhou, China, to investigate the molecular mechanism activated after receiving CoronaVac (Sinovac Biotech Ltd., Beijing, China). Similar to our previous study (Zhang et al., 2022), medical staff who had no fever, cough or other underlying health problems and tested negative for SARS-CoV-2 RNA were eligible for vaccination. The RNA-seq data for healthy control samples (H1, H2, and H3) and patients with SARS-CoV-2 infections (P1, P2, and P3) were downloaded from the Beijing Institute of Genomics (BIG) Data Center (https://bigd.big.ac.cn/), Chinese Academy of Sciences (accession numbers: CRR119890, CRR125445, CRR125446 and CRR119891, CRR119892, CRR119893, respectively). This study was approved by the Ethics Committee of the Sir Run Run Shaw Hospital (Scientific Research 20210210-236), Zhejiang University School of Medicine, Hangzhou, China. Written informed consent was obtained from all participants.

Sampling and PBMC isolation

Venous blood was collected from each medical staff member on Day 21 after the second dose of CoronaVac, centrifuged at 2,000 rpm for ten minutes to obtain serum and stored as aliquots at −80 °C until neutralization antibody detection. PBMCs were isolated using the Ficoll density gradient centrifugation method, frozen in cell-preserving and stored in liquid nitrogen until RNA isolation for transcriptome sequencing.

Detection of SARS-CoV-2 neutralization antibody

The SARS-CoV-2 neutralization antibody in serum was tested using the 2019-nCoV NAb Detection Kit (YHLO, China), which is based on a chemiluminescence immunoassay (Tao et al., 2022). The experiments were carried out according to the instructions of the kit (Liu et al., 2022). The data were statistically analysed using the ROC curve method, and the cut-off value was 10.00 AU/mL. A serum sample from vaccinated staff with a value >=10 was considered reactive.

mRNA library construction and sequencing

Total RNA was extracted from PBMCs using the RNeasy Mini Kit (Qiagen, Hilden, Germany) according to the manufacturer’s instructions. The RNA samples were stored at −80 °C before undergoing an additional DNase I (NEB) treatment. mRNA library construction and sequencing were conducted at GENEWIZ Biotechnology Co, Ltd, Suzhou, China. Briefly, total RNA was used for library preparation. The poly(A) mRNA was purified using oligo (dT) beads. The mRNA was then fragmented into small pieces by divalent cations and high temperature. First-strand cDNA and second-strand cDNA were synthesized. The purified double-stranded cDNA was then treated to repair both ends and add a dA-tail in a single reaction followed by a T-A ligation to add adaptors to both ends. Clean DNA beads were used to select the size of adaptor-ligated DNA. Each sample was then amplified by PCR and validated. The libraries were then sequenced using an Illumina NextSeq 2000 platform according to the manufacturer’s instructions. Quality statistics of raw sequencing data are shown in Table S1.

Data analysis

Identification of differentially expressed genes (DEGs)

The vaccinated individuals, healthy donors and COVID-19 patients had 29, 3 and 3 biological replicates, respectively, when performing RNA sequencing with a sequencing depth of approximately 20X. The statistical power of this experiment was 0.99 as calculated in RNASeqPower (https://doi.org/doi:10.18129/B9.bioc.RNASeqPower). RNA-Seq data for PBMC samples from vaccinated individuals, healthy donors and COVID-19 patients were analysed using the DESeq2 (V1.6.3) Bioconductor package, a model based on the negative binomial distribution. Genes with a p value <0.05 and Log2 >1 for upregulated genes or Log2 <−1 for downregulated genes were considered DEGs; these are shown in the form of heatmaps using SangerBox (http://sangerbox.com/).

Protein −protein interaction (PPI) network construction

The PPI network was constructed using the STRING database (STRING version 11.0; https://stringdb.org/) using 0.9 as the confidence interaction score, and then visualized and analysed with Cytoscape software (3.7.1). The CentiScape and Molecular Complex Detection (MCODE) plugins in the Cytoscape software were applied to screen core modules in the the PPI network to facilitate downstream analysis.

GO and KEGG enrichment analyses

Gene Ontology (GO) and Kyoto Encyclopedia of Genes and Genomes (KEGG) enrichment analyses of DEGs were performed using SangerBox (http://sangerbox.com/). Upregulated and downregulated DEGs were analysed in this study. A p value less than 0.05 was considered a significantly enriched pathway.

Weighted Gene Coexpression Network Analysis (WGCNA)

The coexpression network was constructed using the WGCNA package in R software. In this study, the median absolute deviation (MAD) of each gene was calculated, and the outlier genes were removed with the help of goodSamplesGenes in R software. A β-power = 2 (scale-free R2 = 0.9) was selected as the soft-thresholding parameter to ensure a scale-free network. The adjacency was then transformed into a topological overlap matrix (TOM), which could measure the network connectivity of a gene defined as the sum of its adjacency with all other genes for the network gene ratio, and the corresponding dissimilarity (1-TOM) was calculated. To classify genes with similar expression profiles into modules, average linkage hierarchical clustering was performed according to the topological overlap matrix (TOM)-based dissimilarity measure with a minimum size (gene group) of 30 for the gene dendrogram. To further analyse the module, we calculated the dissimilarity of module eigengenes, chose a cut line for the module dendrogram and merged some modules. A total of 34 modules were identified. The grey module includes genes that cannot be assigned to any module.

Results

Characteristics of participants

A total of 29 participants receiving the second dose of inactivated vaccine were enrolled. The median age of those participants was 28 years (interquartile range, IQR, 22-43) with 58.62% female and participants having a median body mass index (BMI) of 21.61 kg/m2 (IQR, 18.52–24.97) (Table 1).

Table 1 Baseline characteristics of vaccine recipients.

Characteristics	CoronaVac recipients (n = 29)	
Age (year)	28 (22–43)	
Gender		
Female	17 (58.62%)	
Male	12 (41.38%)	
Body mass index (kg/m2)	21.61 (18.52–24.97)	
Notes.

Data expressed as median (IQR) or number (percentage) of vaccine recipients.

SARS-CoV-2 neutralization antibody response elicited by vaccine varied widely

To detect the SARS-CoV-2 neutralization antibody of individuals receiving the second dose of vaccines, the commercialized 2019-nCoV NAb Detection Kit was used. The results showed that all vaccinated staff members had reactivity, but the neutralization antibody response varied considerably among individuals with antibody levels ranging from 10.42 to 489.60 AU/mL (Fig. 1A).

Figure 1 The differentially expressed genes (DEGs) in vaccinated individuals (V), patients (P) and healthy group (H).

(A) The neutralization antibody (NAT) level of vaccinated individuals; DEGs across vaccinated individuals (n = 29), healthy donors (n = 3), and infected patients (n = 3). (B) Volcano plot on DEGs in the comparison of different groups. Red dot represents upregulated DEGs with P-value (two-sided unpaired Wald test) < 0.05 and Log2 > 1.0 and blue indicates downregulated DEGs with P-value < 0.05 and Log2 < −1.0.

Comparative analysis of DEGs

To identify genes that were differentially expressed among vaccinated individuals, COVID-19 patients and healthy controls, the transcriptome data were analysed using the DESeq2 (V1.6.3) Bioconductor package. The genes were screened according to the following standard: differentially expressed genes were those that changed more than two fold and had a q-value <= 0.05. The results showed that gene expression varied widely among the participants (Fig. 1A). The volcano plot showed that 2,184 DEGs were upregulated and 1,765 were downregulated in the vaccinated group compared with the healthy group, whereas 2,844 upregulated DEGs and 3,915 downregulated DEGs were identified in the comparison between the vaccinated group and the patient group (Fig. 1B). When the DEGs of the patient group and healthy group were compared, 1,553 DEGs were upregulated and 1,481 were downregulated (Fig. 1B).

Prioritization of DEGs by PPI network analysis

We further constructed the PPI network of these upregulated and downregulated DEGs using the STRING database and Cytoscape software for visualization with the MCODE k-core decomposition function; 55 upregulated and 10 downregulated DEGs were retained in the vaccinated group compared with the healthy group (Figs. 2A and 2B). Forty upregulated and seven downregulated DEGs were retained in the infection group compared with the healthy group (Figs. 2C and 2D). The upregulated DEGs were mainly associated with the ribosomal subunit, cytokine production, thermogenesis, and the MAPK signalling pathway; downregulated DEGs were related to cell activation, metabolism, and regulation of the inflammatory response. Taken together, we identified many genes involved in the immune response.

Figure 2 Protein–protein interaction (PPI) network of gene signature.

(A) The PPI network extracted from all upregulated DEGs between vaccine and healthy group by MCODE. (B) The PPI network extracted from all downregulated DEGs between vaccine and healthy group by MCODE. (C) The PPI network extracted from all upregulated DEGs between patients and healthy group by MCODE. (D) The PPI network extracted from all downregulated DEGs between patients and healthy group by MCODE.

Functional enrichment analysis of DEGs

To further interpret the transcriptomic changes induced by vaccines in PBMCs with respect to specific cellular biological processes, gene functional enrichment analysis was performed to identify differentially regulated genes and enriched pathways. For the GO terms (BP, CC and MF) of the vaccinated group, the identified DEGs were significantly enriched in “cell activation”, “immune system process”, and “myeloid leukocyte activation” (Fig. 3A and Table S2), which play important roles in the immune response and indicate an activated immune system in PMBCs. For the patient group, the genes were mainly enriched in “complement activation, classical pathway”, “leukocyte mediated immunity”, “humoral immune response mediated by circulating immunoglobulin” and other processes (Fig. 3B and Table S2). Furthermore, the KEGG enrichment analysis indicated that the DEGs in the vaccinated group were related to “ribosome”, “thermogenesis”, “oxidative phosphorylation”, and innate immune response pathways such as “complement and coagulation cascades”, “B-cell receptor signalling pathway”, and “IL-17 signalling pathway” as well as some human disease pathways (Fig. 3C and Table S2). In the patient group, the KEGG pathways were associated with “metabolic pathways”, “lysosome”, “HIF-1 signalling pathway”, and other pathways (Fig. 3D and Table S2).

Figure 3 GO and KEGG enrichment analyses of DEGs in PBMCs transcriptome.

(A) GO enrichment analysis by biological process (BP) and cellular component (CC) between healthy and vaccination group. (B) GO enrichment analysis by biological process (BP) and molecular function (MF) between healthy and patients group. (C) KEGG pathway analysis between healthy and vaccination group. (D) KEGG pathway analysis between healthy and patients group.

WGCNA identified important gene network modules associated with vaccines

We applied the WGCNA method, which has been widely used to identify significant gene modules, to explore the transcriptomic profiling of individuals who received the second dose of inactivated vaccines and infected patients. In the scale-free network, the correlation coefficient was set at 0.9, and the optimal soft-thresholding power was selected as 2 (Fig. 4A). The colours of the dendrogram branches indicate different gene clusters (minimum module size = 30). A total of 34 modules were then screened through hierarchical clustering (Fig. 4B). Based on the association of clinical sample characteristics and the coexpression modules, the module with the strongest correlation with the vaccine was the blue module (R = 0.92, P = 5.3e−15), and this correlation was negative (Fig. 4C). In addition, the blue (R = 0.59, P = 2.1e−4) and green −yellow (R = 0.67, P = 1.0e−5) modules were significantly correlated with SARS-CoV-2 infection status (Fig. 4C). These results indicate that the genes in the blue module were the driving genes of vaccination.

Figure 4 WGCNA of DEGs revealed the gene-network modules.

(A) Scale-free fit index (left) and mean connectivity (right) for various soft-thresholding powers. (B) Clustering dendrogram of DEGs together with assigned module colors. (C) Heatmap of the correlation between modules and clinical traits (including SARS-CoV-2 infection status and vaccination). (D) Cluster of module feature vectors. (E) Blue module.

Analysis of genes in the blue module

To identify the significant and functional genes in the blue module that were strongly related to vaccines, we used Cytoscape software with —MM_R—>0.95 and —GS—>0.9 as the cut-off criteria and removed the points with a degree <3 to create a concentric circle diagram in which six genes (MAPK1, CDC42, PPP2CA, EP300, YWHAZ and NRAS) were defined as the hub genes that had the highest association with vaccines in our study (Fig. 5A). We focused on the genes that were shared between the vaccinated individuals and infected patients in the blue module to explore the molecular mechanism of the protective effect of the inactivated vaccine and selected the top 100 genes that had a significant correlation for subsequent examination, and a new network with the size of the node and the thickness of the edge was developed (Fig. 5B). NRAS, YWHAB, SMARCA5, PPP1CC and CDC5L were selected as candidate genes associated with the inflammatory response and signal transduction. As an inflammatory cytokine that was significantly downregulated in the vaccinated group, NRAS may attenuate the inflammatory response in vaccinated individuals compared with SARS-CoV-2-infected individuals.

Figure 5 Analysis of blue module and identification of genes.

(A) PPI network of genes in blue module of vaccinated group with —MM_R— > 0.95, —GS— >0.9 and removing the points that degree < 3. Node diameter and color intensity represents the degree of connectivity with vaccine. (B) PPI network of top 100 hub genes co-owned in blue modules by vaccinated and patient groups. Node diameter and color intensity represents the degree of connectivity. The thickness of edge indicates the strength of connectivity for each gene.

Discussion

Vaccine strategies are an indispensable tool to defend humans against viruses (Chen et al., 2022b.), including COVID-19 (Zhang et al., 2022), which has resulted in a global epidemic (Fernandes et al., 2022; Wang et al., 2020). It is critical to investigate the molecular mechanisms of the host immune response induced by vaccines at the transcriptional level. In this study, we performed RNA sequencing in PBMCs from 29 individuals immunized with CoronaVac after the second dose and compared these results with the transcriptional data of three healthy controls and three COVID-19-infected patients to characterize the host immune responses and reveal the molecular response mechanism. The main work of this study can be summarized as follows: (I) we systematically analysed the transcriptome profiles of 29 medical staff members receiving an inactivated vaccine compared with three healthy controls and three COVID-19-infected patients; (II) we identified several genes, including MAPK1, CDC42, PPP2CA, EP300, YWHAZ and NRAS that were robustly correlated with the vaccine; (III) NRAS, YWHAB, SMARCA5, PPP1CC and CDC5L were identified as shared response genes, which were significantly downregulated in vaccinated individuals compared with infected patients. These genes may be correlated with the protective effect of inactivated vaccines.

Compared with the healthy controls, the differentially expressed gene functional enrichment analysis in vaccinated groups mainly showed enrichment in the innate immune response, the ribosome, oxidative phosphorylation, thermogenesis, several human diseases and others. In our work, innate immune response pathways mainly included the complement and coagulation cascades, B-cell receptor signalling pathway, IL-17 signalling pathway, and Toll-like receptor signalling pathway (Fig. 3C and Table S2), and the major functional genes in those immune response pathways included TNF, CXCL8, SOS2, IL1R1, JUN, FOS, and IL1B (Table S2). These findings are highly consistent with others in regard to the transcriptional regulation induced by inactivated vaccines. In another study on SARS-CoV-2 inactivated vaccines, a series of innate immune pathways, including the TNF signalling pathway, IL-17 signalling pathway, viral protein interaction with cytokine and cytokine receptor, cytokine −cytokine receptor interaction, and NF-kappa B signalling pathway were observed (Zhang et al., 2021b). In a study of inactivated influenza vaccines, differentially expressed genes associated with the IL-17 signalling pathway and oxidative phosphorylation were detected (Alcorn et al., 2020). For the Hantavax vaccine, post vaccination differentially expressed genes associated with innate immunity and cytokine pathways were highly upregulated (Khan et al., 2019). A previous study showed that the BNT162b2 mRNA vaccine not only stimulated a strong innate immune response such as the Toll-like receptor signalling pathway but also induced epigenetic reprogramming of myeloid cells (Arunachalam et al., 2021). Thus, the innate immune response plays a key rolter vaccination. Recent investigations on COVID-19 vaccines or COVID-19-infected patients also observed enriched pathways related to ribosomes, oxidative phosphorylation and some human diseases (Arunachalam et al., 2021; Forchette, Sebastian & Liu, 2021; Xiong et al., 2020). However, whether the expression of genes in those pathways is associated with the vaccine needs to be further investigated.

To further explore the transcriptomic profiling of inactivated vaccines compared with infected patients, WGCNA was performed, and the results showed that the blue module was significantly negatively correlated with the vaccine. It is essential to determine which genes in the blue module are highly associated with vaccines. As shown in Fig. 5A, NRAS, MAPK1, CDC42, PPP2CA, EP300 and YWHAZ were identified as candidate genes. NRAS acts as an inflammatory cytokine (Chen et al., 2022a) and is related to the TNF-alpha signalling pathway and the IL-17 signalling pathway (Yan et al., 2022). MAPK1 encodes a member of the MAP kinase family, also known as extracellular signal-regulated kinases (ERKs), which acts as an integration point for multiple biochemical signals and is involved in a wide variety of cellular processes such as proliferation, differentiation, transcription regulation and development (Jiang et al., 2021). CDC42 regulates signalling pathways such as the VEGFA-VEGFR2 pathway, which controls diverse cellular functions, including cell morphology, migration, endocytosis and cell cycle progression (Zhou, Wu & Tan, 2022). PPP2CA is one of the four major Ser/Thr phosphatases implicated in the negative control of cell growth and division and consists of a common heteromeric core enzyme associated with a variety of regulatory subunits (He et al., 2019). EP300 functions as a histone acetyltransferase that regulates transcription via chromatin remodelling and is important in the processes of cell proliferation and differentiation. It has also been identified as a coactivator of HIF1A (hypoxia-inducible Factor 1 alpha) and thus plays a role in the stimulation of hypoxia-induced genes such as VEGF (Zhang et al., 2021a). YWHAZ can mediate signal transduction by binding to phosphoserine-containing proteins (Wei et al., 2022). Thus, NRAS and EP300 participate in both metabolic pathways and immune response and regulation, CDC42 is mainly involved in immune response and regulation, MAPK1 and YWHAZ are mainly involved in signal transduction, and PPP2CA is related to metabolism. The correlation between these genes and vaccines may be attributed to cosignalling pathways or the interactions between these genes and vaccines, which need further confirmation.

Aside from revealing the transcriptome profiles of vaccination and screening genes that are strongly associated with vaccines, another goal of this study was to find significantly differentially expressed genes in vaccinated individuals compared with infected patients to explore the molecular mechanism of the protective effect of inactivated vaccines. In the present study, NRAS, YWHAB, SMARCA5, PPP1CC and CDC5L were identified as potential response genes for the protective effect of the vaccine. NRAS is an inflammatory cytokine involved in the development of “cytokine storms” (Chen et al., 2022a) and was significantly downregulated in the vaccinated group in our work. Research has shown that inflammatory cytokine expression correlates with disease severity (Forchette, Sebastian & Liu, 2021; Mehta et al., 2020; Qin et al., 2020). The downregulated inflammatory cytokine NRAS may attenuate the inflammatory response in vaccinated individuals compared with that in SARS-CoV-2-infected individuals and may be one of the molecular mechanisms of the protective effect of inactivated vaccines. YWHAB mediates signal transduction by binding to phosphoserine-containing proteins, which have been shown to interact with RAF1 and CDC25 phosphatases (Kleppe et al., 2014). PPP1CC regulates cellular processes, including cell division (Tchivilev et al., 2008). SMARCA5 has helicase and ATPase activities and regulates the transcription of genes by altering the chromatin structure (Dao et al., 2020). CDC5L is a cell cycle regulator important for the G2/M transition (Qiu et al., 2016). The relationship between the genes of YWHAB, SMARCA5, PPP1CC and CDC5L and the protective effect of the inactivated vaccine needs further study.

There are some limitations to our work. First, raw transcriptome data from healthy controls and patients were downloaded from published papers. Another limitation of this study is that we only analysed the transcriptome data on Day 21 after receiving the second dose of the inactivated vaccine, and thus data for the first and booster doses are lacking. Performing RNA sequencing data at more time points after receiving inactivated vaccines would help provide comprehensive insights into the mechanism of the inactivated COVID-19 vaccine.

Conclusions

In conclusion, our study provides global insights into the transcriptome profiles of host responses in COVID-19-vaccinated individuals and identifies MAPK1, CDC42, PPP2CA, EP300, YWHAZ and NRAS as hub genes that are significantly correlated with the vaccine response. Moreover, NRAS, YWHAB, SMARCA5, PPP1CC and CDC5L were identified as response genes that may explain the molecular mechanism of the protective effect of the inactivated vaccine. Our findings lay the groundwork for the future development of the next generation of vaccines.

Supplemental Information

Table S1 Quality statistics of sequencing raw data

Click here for additional data file.

Table S2 Pathway of GO and KEGG

Click here for additional data file.

Supplemental Information 3 Participant information

The detail information of vaccinated individuals

Click here for additional data file.

We thank all the members for their generous participation.

Additional Information and Declarations

Competing Interests

Author Contributions

Ethics

Data Availability

The authors declare there are no competing interests.

Hongquan Chen conceived and designed the experiments, performed the experiments, analyzed the data, prepared figures and/or tables, authored or reviewed drafts of the article, and approved the final draft.

Lu Zhang performed the experiments, analyzed the data, prepared figures and/or tables, authored or reviewed drafts of the article, and approved the final draft.

Chen Xu analyzed the data, prepared figures and/or tables, authored or reviewed drafts of the article, and approved the final draft.

Xiaoyun Shen performed the experiments, prepared figures and/or tables, and approved the final draft.

Jiazhou Lou performed the experiments, prepared figures and/or tables, data collection, and approved the final draft.

Shengjun Wu conceived and designed the experiments, authored or reviewed drafts of the article, and approved the final draft.

The following information was supplied relating to ethical approvals (i.e., approving body and any reference numbers):

The Medical Ethical Committee of Sir Run Run Shaw hospital

The following information was supplied regarding data availability:

The data is available at the National Genomics Data Center GSA-Human, accession number: HRA003858.

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
