# Peer review of "Analysing transcriptomic signatures and identifying potential genes for the protective effect of inactivated COVID-19 vaccines"

_PeerJ, doi:10.7717/peerj.15155_

## Round 0.1 · original submission · Minor Revisions

Please revise the manuscript as the reviewers suggested.

Reviewer 1 ·

Basic reporting

In this manuscript, Chen et al. analyzed neutralization antibody responses elicited by vaccine serum and carried out transcriptome sequencing of the RNAs isolated from PBMC of 29 medical staff receiving two doses of CoronaVac vaccines. The authors found that SARS-CoV-2 neutralization antibody titers varied considerably among individuals, and revealed that many innate immune pathways were activated after vaccination. After analyzing many genes, the authors found that NRAS, YWHAB, SMARCA5, PPP1CC and CDC5L may be correlated with the protective effect of inactivated vaccine, while MAPK1, CDC42, PPP2CA, EP300, YWHAZ and NRAS were demonstrated as the hub genes that had significant association with vaccines. This study provides a resource for understanding the molecular mechanism of host immune response induced by inactivated vaccine. This study is important and timely to the field and I suggest to publish this study after minor revision.

Minor points
1) The resolution of the main figure needs to be improved.

2) The English language is a bit hard to understand and the authors are suggested to invite fluent English speakers to edit the manuscript.

Experimental design

Great

Validity of the findings

Great

Reviewer 2 ·

Basic reporting

1. The language and grammar should be improved. I list some below and strongly suggest the authors ask a fluent English-speaking colleague or a professional editing service to polish the manuscript.
Line 84, has caused
Line 85, have been
Line 93, were developed
Line 114, a published paper
Line 115, Differential gene expression analysis
Line 146, neutralization antibody in serum was tested
Line 216, was
Line 218, staffs

2. More descriptions and explanations should be added after PPI Network analysis and GO analysis.
3. I suggest the authors move one or two genes’ function descriptions from the Discussion to the Result section, to describe or explain the relationship between there hub genes and vaccination.
4. Lines 237-238, it should be “Fig. 2A and 2B”, and “Fig. 2C and 2D”.

Experimental design

no comment

Validity of the findings

no comment

Additional comments

1. Did authors remove the batch effect between different source datasets?
2. I suggest the authors verify the expression of critical genes by quantitative real-time PCR in the enrolled participants. Of course, it will improve the manuscript if these genes could be verified in more vaccinated individuals.

·

Basic reporting

1.Line 107 to Line 109
In the Introduction, I don't think there's enough of an argument here. What are the other transcriptomic results of patients infected by SARS-CoV-2 or other strains like it? What are the innovations of this study? I suggest that these statements be added to these paragraphs in Introduction.
2.Line 176 ... and |Log2 (fold change)| > 1 ...
Line 177 ... |Log2 (fold change)| < -1 ...
In this case, it is recommended to use the subscript for the 2 of Log2. In addition, authors should revise the similar typos carefully.

Experimental design

1.Line 146 to Line 147
The SARS-CoV-2 neutralization antibody in serum were tested using 2019-nCoV Nab Detection Kit (REF C86109, YHLO, Shenzhen, Guangdong, China).
This kit could not be found on the official website of this company, so I suggest you provide the relevant link.
2.Whether is the sample number of patients and healthy donors too small?

Validity of the findings

1.Line 215 to Line 219, and Figure 1
Why the Healthy 1 group is closer to the Patient 1 group in cluster analysis of Figure 1? The authors are suggested to explain and revised.

And most importantly, did the study take into account potential differences between differential vaccine providers? If possible, the authors are invited to discuss in combination with a large number of current clinical studies (cited references).

In addition, to adjust the clarity of the figures, it is recommended to use vector format. Then, this will allow readers to better understand the manuscript.

Additional comments

This study analyzed the transcriptome of individuals vaccinated with COVID-19 vaccine and revealed key genes related to immune response and protective effect. The detailed analysis has laid a foundation for understanding the molecular mechanism of immune response induced by inactivated vaccine and subsequent vaccine development research. This manuscript meets the aims and scopes of Peer J, and I suggest that this manuscript might be published after minor revision.

---

## Round 0.2 · accepted · Accept

This study can be accepted now.